



# Wintertime carbon uptake of managed temperate grassland ecosystems may influence grassland dynamics

Genki Katata[1,2], Matthias Mauder[2], Matthias J. Zeeman[2], Rüdiger Grote[2], and Masakazu Ota[3]

[1]Institute for Global Change Adaptation Science (ICAS), Ibaraki University, Ibaraki, 310-8512, Japan
[2]Institute of Meteorology and Climate Research, Atmospheric Environmental Research (IMK-IFU), Karlsruhe Institute of Technology, Garmisch-Partenkirchen, 82467, Germany
[3]Research Group for Environmental Science, Japan Atomic Energy Agency (JAEA), Ibaraki, 319-1195, Japan

**Correspondence:** Genki Katata (genki.katata.mirai@vc.ibaraki.ac.jp)

**Abstract.** Rising temperatures and changes in snow cover, as can be expected under a global warmer climate, may have large impacts on mountain grassland productivity limited by cold and long winters. Here, we elaborated a multi-layer atmosphere-soil-vegetation model to account for snow, freeze-thaw events, grass growth, and soil microbiology. The model was applied to simulate the responses of managed grasslands to anomalously warm winter conditions. The grass growth module represented

key ecological processes under a cold environment, such as leaf formation, elongation and death, tillering, carbon allocation, and cold acclimation, in terms of photosynthetic activity. Input parameters were derived for the pre-alpine grassland sites in Germany, for which the model was run using three years of data that included a winter with an exceptionally limited amount of snow cover. The model reproduced the temporal variability of observed daily mean heat fluxes, soil temperatures and snow depth throughout the simulation period. High physiological activity levels during the extremely warm winter led to a simulated

$CO_2$ uptake of 100 gC m$^{-2}$, which was mainly allocated into the below-ground biomass and only to a minor extend used for additional plant growth during early spring. If this temporary dynamics is representative of the long-term changes, this process, which is so far largely unaccounted for in scenario analysis using global terrestrial biosphere models, may lead to carbon accumulation in the soil and/or carbon loss from the soil as a response to global warming.

## 1 Introduction

Grassland's productivity in temperate and boreal regions is important for food production as a means of fodder for livestock, and is expected to be highly influenced by climate change (Jing et al., 2014; Tubiello et al., 2007). It is also expected that mountain grassland ecosystems are particularly sensitive under climate warming scenarios, with future changes of snow cover at high altitudes (Xie et al., 2017). Therefore, understanding the response of mountain grassland productivity to snow cover conditions is crucial for the future prediction of grassland-based food productivity as well as possible feedback of carbon and

energy balances in grassland ecosystems to climate change.

Although forage production from grasslands is known to be limited by cold and long winters in mountainous regions, there are still uncertainties regarding winter stresses on grassland vegetation (e.g., grasses, clover, other herbaceous species, flowers,





and mosses) under a future climate (Rapacz et al., 2014). The properties of winter stress are complex, depending not only on environmental factors such as low temperature during winter, but also largely on the presence or absence of snow cover

and factors that control the acclimation status of grassland vegetation to cold (Ergon et al., 2018). For example, winter stress due to a low temperature limits the productivity of grassland vegetation, as it is linked to dormancy of grassland vegetation in cooler regions and at higher elevations (i.e., with lower ambient temperature) being dormant during winter. On the other hand, as for the effects of snow cover, a shorter duration of snow period was observed in a recent observational study at upland temperate grasslands (Zeeman et al., 2017), showing that grasslands at the low-elevation sites with a short snow period

are photosynthetically active throughout the winter, while grassland vegetation remains dormant at sites of higher elevation even under the snow-free conditions. As a result, gross primary production (GPP) drastically increased at the low-elevation grasslands during the snow-free winter, enabling rapid spring growth that is mainly driven by soil temperature (Zeeman et al. 2017). Clearly, it is necessary to understand the response of grassland productivity to the above change in snow cover conditions in order to accurately estimate the response of the carbon cycle in mountain grassland ecosystems to future climate

change.

Winter stress influences the carbon dynamics in grassland vegetation in the growing season. The underlying mechanism of winter stress is that photosynthesis continues during winter in frost-tolerant species (Höglind et al., 2011; Tuba et al., 2008), but growth stops if soil temperatures are lower than 5 °C (Körner, 2008). In this situation, organic matter (organic carbon) produced by photosynthesis is not used for grass growth but accumulates in the plant as reserves during winter (e.g.,

Körner, 2008). The sink-limitation processes due to cold temperatures or any sink-limitation on growth is not accounted for in current grassland models. However, its importance increases under climate change since photosynthetic conditions may improve particularly during winter and the onset of spring growth may occur earlier (e.g., Desai et al., 2015). Therefore, the importance of representing wintertime grassland productivity considering direct and indirect impacts of climate (e.g., snow cover and its impact on soil temperatures) needs to be addressed.

This research focuses on how temperate grassland productivity responds to temperature and snow cover duration in mountainous areas. The underlying hypothesis is that winter dynamics is important for mountainous ecosystem carbon balance although most existing grassland models for temperate climate conditions focus exclusively on the spring and summer growing season (Höglind et al., 2016). In particular, sink limitations for grassland vegetation growth limited by environmental (e.g., temperature, water, and nutrient controls) or plant internal (e.g., ontogenetic) factors other than $CO_2$ assimilation (Fatichi et

al., 2019; Körner et al., 2007) are not included as suggested by recent studies (Fatichi et al., 2014; Van Oijen et al., 2018). Therefore, we suggest a process-based land surface model that can simulate both physical (snow and freeze-thaw) and biological processes (carbon allocation under cold stresses) and includes these sink limitations. This new model is developed based on a multi-layer atmosphere-SOiL-VEGetation model (SOLVEG; Katata et al., 2014), and is applied to the $CO_2$ flux sites at two managed grasslands in the German pre-alpine region over a number of years that featured normal (2011-2012 and 2012-2013)

as well as extremely warm (2013-2014) winters (Zeeman et al., 2017). The results are evaluated with measurements and are discussed based on sensitivity analysis.





## 2  Materials and Methods

### 2.1  SOLVEG

A one-dimensional multi-layer model SOLVEG consists of four sub-models: atmosphere, soil, vegetation, and radiation within

the vegetation canopy as shown in Fig. 1. The general description is available in Katata (2009), Katata and Ota (2017), Nagai (2004), and Ota et al. (2013). Details of the processes of snow accumulation and melting, freeze-thaw in soil, and grassland vegetation growth and development are described in the supporting information.

In the atmosphere sub-model, one-dimensional diffusion equations are solved between atmospheric layers for horizontal wind speeds, potential temperature, specific humidity, liquid water content of the fog, turbulent kinetic energy and length

scale (Katata, 2009), and gas and aerosol concentrations (Katata and Ota, 2017). Observational data are used for the upper boundary conditions. Bulk transfer equations are applied at the lowest layer using the soil surface temperature and specific humidity calculated in the soil sub-model. In the soil sub-model, the soil temperature, volumetric soil water content, and specific humidity in the soil pores are predicted based on heat conduction, mass balance in liquid water, and water vapor diffusion equations, respectively (Katata, 2009). Root water uptake is calculated from the transpiration rate in the vegetation

sub-model. For $CO_2$ concentration in soil, mass conservation equations for liquid and gas phases are solved (Nagai, 2004). Organic matter dynamics are also considered (Ota et al., 2013) as microbial decomposition and dissolved organic carbon (DOC) leaching in the above-ground litter layer, below-ground input of carbon from roots (root litter), and soil organic carbon (SOC) turnover and DOC transport along water flows throughout the soil profile for three SOC pools (active, slow, and passive) with different turnover times.

In the vegetation sub-model, profiles of the leaf temperature, leaf surface water, and the vertical liquid water flux were predicted (Nagai, 2004). The heat budget equation at the leaf surface is solved to predict the leaf temperature using key variables from the atmosphere sub-model combined with the radiation scheme. At the upper boundary of the sub-model, the given precipitation intensity is used for calculating vertical liquid water flux within the canopy based on the surface water budget equation. The $CO_2$ assimilation rate due to photosynthesis is predicted using the Farquhar's formulations (Farquhar et

al., 1980) and stomatal resistance. In the radiation sub-model, direct and diffuse downward and upward fluxes of solar and long-wave radiation are calculated to obtain the radiation energy input at the canopy layers. Fractions of sunlit and shaded leaves at each canopy layer are computed for the stomatal resistance and energy budget calculations.

A multi-layer snow module is mainly developed based on either the Community Land Model (CLM; Oleson et al., 2010) and SNTHERM (Jordan, 1991), while the model is unique in including the gravitational and capillary liquid water flows in the

unsaturated snow layer based on van Genuchten's concept of water flow in the unsaturated zone (c.f., Hirashima et al., 2010). In the soil module, freeze-thaw processes in soil based on the freezing-point depression equation (Zhang, Sun, and Xue, 2007) are considered in heat conduction and liquid water flow equations.

To simulate the winter-related processes for grassland phenology such as leaf development and senescence due to cold stresses, the relevant scheme in the grass growth model named BASic GRAssland model (BASGRA; Höglind et al., 2016) is

coupled with the vegetation sub-model of SOLVEG to simulate vegetation growth. The three main features that characterize





plant growth in BASGRA are: (1) simulation of source-sink relations where the source consists both current photosynthesis and remobilization of reserves; (2) simulation of leaf area dynamics and tillering for vegetative and generative tillers; and (3) cold hardening and the effect of physical winter stress factors on tiller survival and plant growth. BASGRA has been well validated by using several experimental datasets of harvestable dry matter of perennial rye grass collected in Europe (Schapendonk et al., 1998) and from five locations in Norway, covering a wide range of agroclimatic regions, day lengths, and soil conditions (Höglind et al., 2016). BASGRA consists of the LINGRA grassland model (Van Oijen et al., 2005) with models for cold hardening and soil physical winter processes, while diurnal $CO_2$ assimilation is calculated as accumulation of the net assimilation for each time step within the vegetation sub-model (Nagai, 2004) instead of the original scheme of photosynthetic processes in BASGRA. When snow covers grasses, no photosynthesis is assumed to occur due to low light availability and only soil respiration is considered. BASGRA uses a so-called "big-leaf" approach (Monteith, 1981), thus predicting the total LAI of the whole grassland vegetation canopies. Since SOLVEG uses a multi-layer structure of canopies, the profile of leaf area density is obtained from simply dividing total LAI by canopy height (h) by assuming vertically uniformity for all canopy layers. Canopy height, which is not simulated in BASGRA, is calculated by the function of LAI with fitting parameters.

Carbon gain from photosynthesis and remobilized reserves are allocated among sinks based on changing sink priorities and strengths. Sink strengths are calculated based on the dynamics of leaves and stems and the acclimation to low temperature. The following five sinks are considered: the processes of cold hardening, replenishment of the reserves pool, leaf growth, stem growth, and root growth. Sink strengths are defined as the rate at which these processes would proceed with no source limitation. The hardening process has top priority, so its demand is met in full if source strength is large enough, irrespective of the four other sinks. Root growth has lowest priority and depends on carbon unused by other sinks. The strength priority between reserves on the one hand, and leaves and stems on the other hand changes with day length. When day lengths are shorter than a cultivar-specific threshold, reserves have higher priority than stems and leaves, with the opposite during the rest of the year. Leaves and stems have equal priority so they receive carbon according to their sink strengths. The removal of tillers and leaves by cutting can be simulated during the growing season, with subsequent regrowth of the sward. The regrowth rate after cutting is calculated at each phenological stage. Natural turnover of leaves and roots is modeled using typical life spans in years (Arora and Boer, 2005), while BASGRA does not simulate the senescence of elongating tillers or roots. The fraction of roots in soil layers and rooting depth are modeled as a function of root biomass (Arora and Boer, 2003), which may be required to be tested at multiple biomes. Daily amounts of the dead root biomass (root litter) are used as inputs to SOC in the soil sub-model of SOLVEG.

## 2.2 Empirical parameterizations for cold acclimation

Although the relation between the maximum catalytic capacity of Rubisco ($V_{cmax}$) and air temperature is quite well established (e.g., Bernacchi et al., 2001; Leuning, 2002; Smith and Dukes, 2013), parameters related to photosynthesis are still uncertain (Kattge and Knorr, 2007) also for low temperature (Höglind et al., 2011). Thus, in the vegetation sub-model, we introduced the empirical factor for cold stress of grasslands, $f_{cold}$, to empirically simulate the reduction of photosynthesis under low air temperature as per the following equations (see also Supplement):





$$A_n = min(f_{cold}w_c, w_e, f_{cold}w_s) - R_d, \tag{1}$$

$$f_{cold} = \min\left[1, \max\left\{0, \frac{(T_a + 4)}{(T_{ph} + 4)}\right\}\right], \tag{2}$$

where $A_n$ ($\mu$mol m$^{-2}$ s$^{-1}$) is the net CO$_2$ assimilation rate at each canopy layer, which is calculated by subtracting the leaf respiration rate $R_d$ ($\mu$mol m$^{-2}$ s$^{-1}$) from the assimilation rate, $w_c$ ($\mu$mol m$^{-2}$ s$^{-1}$) is the limitation by efficiency of the photosynthetic enzyme system (Rubisco), $w_e$ ($\mu$mol m$^{-2}$ s$^{-1}$) is the limitation by the absorbed photosynthetically active

radiation (PAR), $w_s$ ($\mu$mol m$^{-2}$ s$^{-1}$) is the limitation by the capacity of leaf to export the products of photosynthesis, $T_a$ (°C) is the daily and vertical mean air temperature for all canopy layer, and $T_{ph}$ (°C) is the threshold air temperature above which grasslands are photosynthetically active. Determination of the value of this threshold temperature is important to avoid the overestimation (mainly from fall to winter) of photosynthesis at a low temperature (Höglind et al., 2011). In the original BASGRA, $T_{ph}$ is set to 1 °C, that is, $V_{cmax}$ starts decreasing linearly when $T_a$ drops below 1 °C until it becomes zero at $-4$°C.

However, in the SOLVEG simulation, since the values of $T_{ph}$ may change depending on environmental conditions, the value of $T_{ph}$ is calibrated for each site so that the model reproduce the observed CO$_2$ flux during the extremely warm winter period.

## 2.3 Study sites and observational data

The model is applied to two sites of managed grassland named the Graswang (47.5708 °N, 11.0326 °E, 864 m asl.) and the Fendt (47.8329 °N, 11.0607 °E, 595 m asl.) belonging to the TERestrial ENvironmental Observatories (TERENO) network

in Germany. General information on the climate and management of sites is available in Table 1. Both sites are located in the Bavarian Alpine Foreland, which is an area in the south of Germany and north of the Alps (Mauder et al., 2013; Zeeman et al., 2017; Zeeman et al., 2019). The grasses are harvested several times during the growing season defined as the period from April to October.

Half-hourly data of precipitation, atmospheric pressure, horizontal wind speed, air temperature and humidity, and incoming

long- and short-wave radiation were used at the top atmospheric layer as a height of 3.5 m. Data of friction velocity ($u*$), sensible ($H$) and latent heat ($\lambda E$), and CO$_2$ fluxes ($F_{CO2}$) observed over the grassland based on the open-path eddy covariance method using a three-dimensional sonic anemometer (CSAT3; Campbell Scientific, USA) and an open-path CO$_2$/H2O gas analyzer (LI-7500; Li-Cor, USA) were used for validation of the simulation results. The net radiation ($R_{net}$) over the canopies, soil temperature at 0.05 m in depth, and snow depth were also used to evaluate the simulated surface energy and water balances.

Details of the site characteristics and micrometeorological observations are described by Zeeman et al. (2017).

## 2.4 Calibration and validation procedure

Parameters used for SOLVEG simulations are summarized in Table 2. The simulation period is approximately three years from 1 December, 2011, to 1 November, 2014, which included both normal (2011-2012 and 2012-2013) and extremely warm (2013-2014) winters. Since the lack of the data, most of the micrometeorological and hydrological parameters for SOLVEG





runs are from previous studies conducted at the study sites (Hingerl et al., 2016; Kunstmann et al., 2004; Kunstmann et al., 2006). The set of parameters of BASGRA for typical perennial grass species of timothy in the Nordic region (Höglind et al., 2016) is applied. Grass cutting events are determined from clear reductions in $CO_2$ flux, surface albedo and phenology camera observations according to Zeeman et al. (2017). The unknown parameter, the threshold temperature for cold stresses [$T_{ph}$ in Eq. (2)], is manually determined in the simulation for each site to obtain a good agreement between simulated and measured

$CO_2$ flux over the canopy during winter. The calibration results of daily mean surface fluxes ($R_{net}$, $H$, $\lambda E$, and $F_{CO2}$) are statistically evaluated using the mean error (ME), the root mean squared error (RMSE), intercept and slope of linear regression lines, and the Pearson's correlation coefficient (R).

### 2.5 Scenario determination for sensitivity analysis

To investigate the impact of cold acclimation of grassland vegetation on the $CO_2$ balance and carbon allocation at mountain

grassland ecosystems, two scenarios using the SOLVEG model are defined based on the experimental results of Höglind et al. (2011): "active scenario" ($T_{ph}$ = 1 °C) and "dormant scenario" ($T_{ph}$ = 11 °C). The former indicates that photosynthesis is active during most of the wintertime and photosynthesis works even at the low temperature of 1 °C. In contrast, the latter represents the situation where grass physiology is more or less shut down and photosynthesis ceases under the condition of a relatively high temperature of 11 °C to protect from cold death. Both scenarios are adopted for both the Graswang and Fendt

for the same period.

## 3 Results

### 3.1 Model calibration and validation

Figure 2 shows the temporal changes in simulated and observed daily surface heat fluxes over the grassland at the Fendt and Graswang throughout the three-year simulation period. The model generally reproduced the typical seasonal changes measured

at both sites, for example, low values of the Bowen ratio ($H/\lambda E$) at the Fendt during the growing season (from April to October) and negative sensible heat flux ($H$) at the Fendt in December 2013, as suggested by Zeeman et al. (2017).

Figure 3 illustrates the time series of modelled and observed daily soil temperature and snow depth at the two sites. Observed changes in snow depth were reproduced by the model overall (Fig. 3a, c). Seasonal changes in observed soil temperature were also reproduced by the model; for example, when the grassland was under the snow cover at the Graswang from December

2012 to February 2013, soil temperature at a depth of 0.02 m remained almost 0 °C for both observed and simulated values (Fig. 3c). Sudden increases in soil temperature over the snow-free condition were also reproduced by the model; this was particularly evident at the Fendt during the extremely warm winter of 2013-2014 (Fig. 3a).

Simulated and observed daily $CO_2$ fluxes ($F_{CO2}$) over the canopies and simulated LAI at both sites are presented in Fig. 3. The model simulated the observed increase of $CO_2$ flux after the harvesting, which was achieved by the regrowth of grassland

vegetation (Fig. 3b and d). During the extremely warm winter from December 2013 to February 2014, negative values of





observed $CO_2$ flux at the Fendt were reproduced by the model (Fig. 3b) using the calibrated value of $T_{ph}$ = 1 °C (Table 1). At the Graswang, both observed and simulated $CO_2$ fluxes were very small and near to zero (Fig. 3d) due to a high threshold temperature for cold acclimation calibrated as $T_{ph}$ = 11 °C (Table 1).

Scatter diagrams and statistical comparisons of daily energy and $CO_2$ fluxes at the two sites throughout the simulation period are presented in Fig. 4 and Table 3, respectively. At both sites, the slopes of the regression lines were overall close to unity and values of the intercepts were sufficiently small for $R_{net}$, $H$, and $\lambda E$. High correlations were also observed between measured and simulated $CO_2$ fluxes at both sites.

## 3.2  Sensitivity analysis

Figure 5 illustrates temporal changes in simulated snow depth and leaf biomass obtained for the active and dormant scenarios for the normal winter (2012-2013) and extremely warm winter (2013-2014). Significant differences between the two scenarios of at most a factor of two were found in the results during winter. Nevertheless, the leaf biomasses at the first cutting event from May to June were similar at both sites and scenarios.

Figure 6 depicts the selected results of cumulative GPP and ecosystem respiration (RE), and mean leaf and root biomasses, carbon reserve content (total stock of carbon that can be allocated to any of the plant elements such as leaves, stems, and roots), and LAI simulated at the Fendt during winter and spring in 2014. Both GPP and RE were higher in the active scenario than in the dormant one as expected by the model construction (Fig. 6a and b); this was particularly apparent as GPP differed by a factor of three or by approximately 100 gC m$^{-2}$ (Fig. 6a). Nevertheless, changes in leaf biomass and LAI during the subsequent spring in the active scenario were clearly lower than in the dormant scenario (Fig. 6c and f). In contrast, changes in root (below-ground) biomass during spring in the active scenario were approximately three times higher than in the dormant scenario (Fig. 6d). Simulated carbon reserve contents in both winter and spring were similar in the two simulation scenarios (Fig. 6e) because the carbon fixed by photosynthesis was immediately allocated to the above- or below-ground biomass.

## 4  Discussion

The results demonstrate that the modified SOLVEG model that includes the physical (snow and freeze-thaw) and biological processes (carbon allocation under cold stresses) based on the existing land surface model (SOLVEG) can reasonably simulate heat and carbon transfer processes in managed grassland ecosystems (Figs. 2-4 and Table 3). In particular, the model reproduced the low or near-zero $CO_2$ uptake during the normal winter period at the Graswang, regulated by the lowering of soil temperature due to snowfall (Fig. 3d). On the other hand, the observed high uptake of $CO_2$ at the Fendt in the extremely warm winter was also simulated by the model (Fig. 3b). The key parameter that determined the above $CO_2$ uptake processes was the threshold air temperature of $T_{ph}$ in Eq. (2) for the photosynthetic activity level of grassland ecosystems. Tuning of the above parameter is required for each site to simulate carbon dynamics in the grassland ecosystems in cold climate regions.

Our approach uses the manually calibrated $T_{ph}$ values for each site, while only typical (average) values are taken for different plant functional types of grassland vegetation in global biogeochemical models. Numerical experiments using $T_{ph}$ = 1 °C





revealed that the high $CO_2$ uptake rate at low altitude during winter was likely explained by high levels of physiological activity of grasslands (Fig. 5a). In this experiment, the impact of cold acclimation on the $CO_2$ balance for the two pre-alpine

temperate grassland sites was evaluated by manually tuning the threshold temperature of photosynthesis to lower ($T_{ph} = 1$ °C) and higher values ($T_{ph} = 11$ °C) because the exact mechanism of model response to $T_{ph}$ changes is unclear (Höglind et al., 2011). A possible explanation for the altered photosynthesis is the exposure to freezing temperatures since this leads to rapid acclimation responses (e.g., Huner et al., 1993; Kolari et al., 2007). The study site was exposed to frost during the extremely warm winter in 2013-2014 (Zeeman et al., 2017), for which a rapid acclimation is a likely reason for the observed decline in

photosynthetic capacity. Thus, this study is still the preliminary step to define minimum thresholds of photosynthesis in order to model this process for various grassland ecosystems.

The high $CO_2$ uptake rate during the snow-free conditions was not limited to the Fendt site, but is likely a wide-spread phenomenon at other mountain grasslands in Europe. This is illustrated in Table 1, which summarizes the full-year observational studies that include wintertime $CO_2$ flux at European mountains. Indeed, except for the Austrian site of Rotholz, which has a

long grazing period that may intensively reduce grass productivity (Wohlfahrt et al., 2010), high $CO_2$ uptake during snow-free periods was observed at all altitudes below 760 m, corresponding to annual mean air temperature (MAT) of less than 8 °C. If the altitude or MAT is considered as a threshold of cold acclimation of grasses, the snow-free wintertime $CO_2$ uptake may have a large impact on the carbon balance of grassland ecosystems over the European Alps. Since a rise of snowline and wintertime air temperature up to 300-600 m or 2-4 °C, respectively, has been predicted for the latter part of the 21st century, the effect is

even likely to increase (Gobiet et al., 2014). It should be noted, however, that other indicators of the level of cold acclimation might be superior to the use of MAT because physiological activities of grassland vegetation are often triggered by temperatures during specific development stages. If, however, such activities are rather closely related to the MAT (as indicated in Table 1), it is also possible that the differences in phenology and photosynthesis are caused by a different species composition of grasslands. In this case, the acclimation speed and management options that facilitate a change to better adapted ecosystems

should be investigated.

Using the modified SOLVEG model that considers carbon dynamics in grassland vegetation that depends on the source strength and the sink demands, the results of the active scenario demonstrated that a large fraction of carbon ($CO_2$) gained by photosynthesis during winter was not directly allocated into above-ground biomass but used to grow roots during the spring growth period (Fig. 6d), currently unsupported by observations. Most studies of alpine grassland ecosystems in Europe have

focused on the impact of climate changes on grass yield (i.e., grassland-based food production); for example, in the Nordic region, future $CO_2$ increase, warming, and less snowfall are expected to increase the grassland productivity (Ergon et al., 2018). According to this study, $CO_2$ uptake at the Fendt in the extremely warm winter increased annual GPP to 100 gC m$^{-2}$ due to cold acclimation in the active scenario, but the above-ground biomass of the first cutting simulated in this scenario was less than that in the dormant scenario (Fig. 6c). This indicates that grass yield cannot be simply determined by the source-strength

($CO_2$ assimilation due to photosynthesis) and is controlled by the sink-demand of the above-ground biomass (foliar, tiller, and stem growth). Indeed, an open-top-chamber warming experiment in the alpine steppe on the north Tibetan Plateau showed that warming significantly increased total root biomass by 28 % at a soil depth of 0-0.01 m in the growing season (Ma et al.,



2016), supporting the possibility of larger below-ground allocation of organic carbon, as suggested by this study. Therefore, the increased photosynthesis in the warmer winter does not necessarily increase grass yields. Further experimental evidence

linking between the above- and below-ground processes is required to obtain an accurate understanding of the carbon dynamics of grassland ecosystems.

Another important implication from numerical experiments is that carbon stock/loss in/from the soil in the mountain grasslands may be greater in a future warmer climate. The root biomass simulated for the active scenario was three times greater than that for the dormant scenario (Fig. 6d), indicating that more carbon is accumulated in the soil by root death (root litter

input) in grassland ecosystems in warmer winters. Indeed, recent studies suggest that a relatively high MAT accelerates the turnover of roots to produce root litter input in managed mountain grassland ecosystems (Leifeld et al., 2015). This change in the below-ground input of carbon in grassland ecosystem is particularly important for the carbon cycle at managed grassland ecosystems because plant-fixed carbon from the above-ground biomass is substantially reduced following a cut. Furthermore, this may enhance carbon loss from the soil due to heterotrophic respiration and leaching of $CO_2$ because grassland vegetation

typically has a high density of fine roots that are poorly lignified and with high turnover rates, providing a relatively labile carbon substrate for microbial activity (Garcia-Pausas et al., 2017). The altered SOC dynamics in grassland ecosystems may be of considerable importance for the global carbon cycle since soils of temperate grassland ecosystems are already estimated to hold a large stock of carbon, that is, 7 % of total global soil carbon (Jobbágy and Jackson, 2000). Therefore, we suggest that global terrestrial biosphere models (Fatichi et al., 2019) need to be elaborated with phenological and acclimation processes as

interactions with below-ground processes (Gill et al., 2002; Riedo et al., 1998; Soussana et al., 2012) in order to estimate the carbon balance response of managed grassland ecosystems to global warming.

*Data availability.* The output data in this study are publicly accessible via contacting the first author.

*Author contributions.* GK developed the model with supports from RG and MO, and performed the simulations using the data collected by MM and ZM. GK prepared the manuscript with contributions from all co-authors.

*Competing interests.* We have no conflict of interest to declare.

*Acknowledgements.* We thank the staff of KIT/IMK-IFU in Germany and ICAS in Japan for their support. We also express our gratitude to Dr. Georg Wohlfahrt of the University of Innsbruck, Austria; Dr. Jun Koarashi of the JAEA, Japan; Dr. Kentaro Takagi of Hokkaido University, Japan; and Dr. Ankur Desai of Wisconsin University for their helpful comments and suggestions on this study. The German weather data were provided by DWD. Fortran code of BASGRA was provided from http://dx.doi.org/10.5281/zenodo.27867. Output data in

this study are all publicly available and are included in the supporting information. The TERENO pre-Alpine infrastructure is funded by the





Helmholtz Association and the Federal Ministry of Education and Research. One of our co-authors, Dr. Matthias Zeeman, received support from the German Research Foundation (DFG; grant number ZE1006/2-1). This study was partly supported by a Postdoctoral Fellowship for Research Abroad and a Grant-in-Aid for Scientific Research (No. 22248026), and Leading Initiative for Excellent Young Researchers, provided by the Japan Society for the Promotion of Science and the Ministry of Education, Culture, Sports, Science and Technology.



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



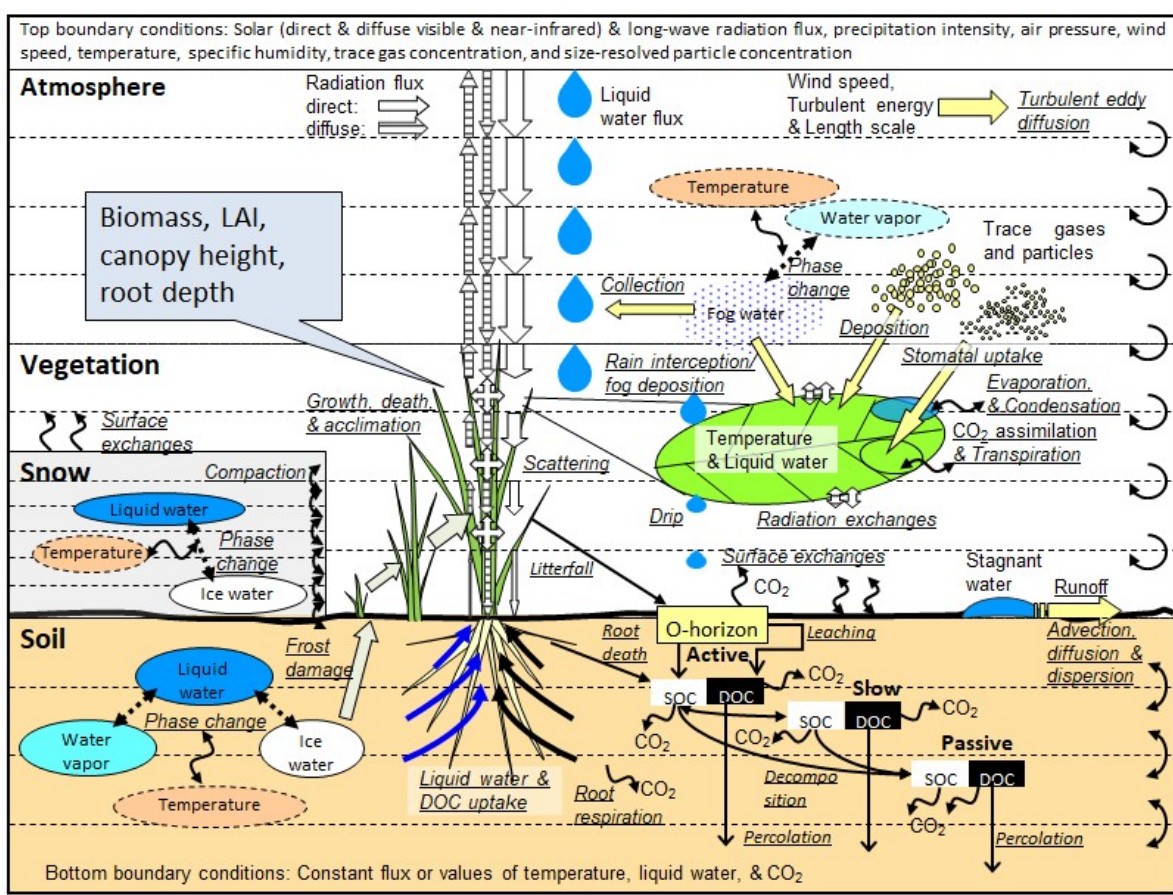

**Figure 1.** Overview of biological and physico-chemical variables and processes (underlined words) for atmosphere, soil, vegetation, and snow submodels in SOLVEG. The part of the existing grass growth model of BASGRA is coupled in this study.



**Figure 2.** Time series for (a, c) calculated (lines) and (b, d) observed (open circles) daily mean net radiation ($R_{net}$), sensible heat flux ($H$), and latent heat flux ($\lambda E$) at (a-b) Fendt and (c-d) Graswang throughout the calculation period.





**Figure 3.** Time series for calculated (solid lines) and observed (open symbols) (a, c) daily mean soil temperature at a depth of 0.02 m, snow depth, and (b, d) $CO_2$ flux ($F_{CO2}$), and leaf area index (LAI) at (a-d) Fendt and (e-h) Graswang throughout the calculation period. Sudden decreases in calculated LAI in (b, d) represent grass cutting events.

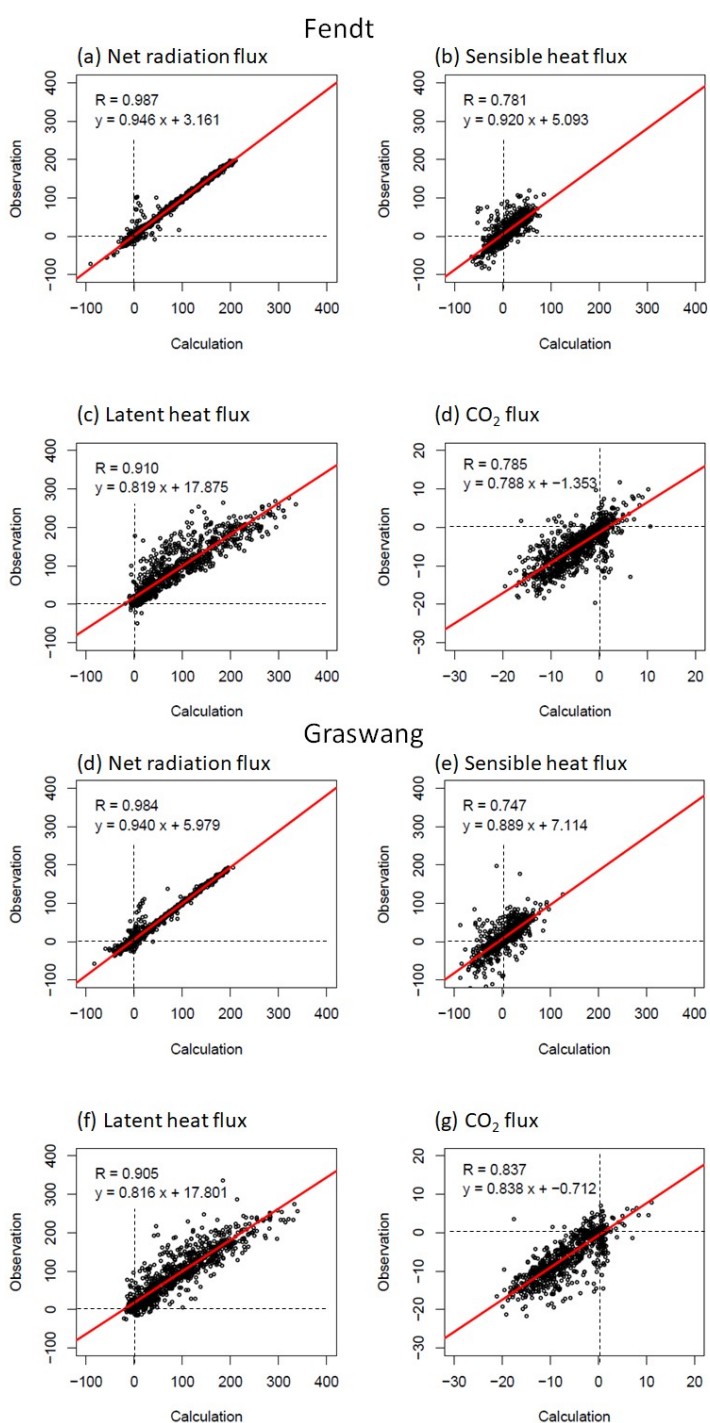

**Figure 4.** Scatter diagrams of calculated and observed (a, e) daily mean net radiation ($R_{net}$), (b, f) sensible ($H$) and (c, g) latent ($\lambda E$) heat, and (d, h) $CO_2$ fluxes ($F_{CO2}$) at (a-d) Fendt and (e-h) Graswang for the calculation period.

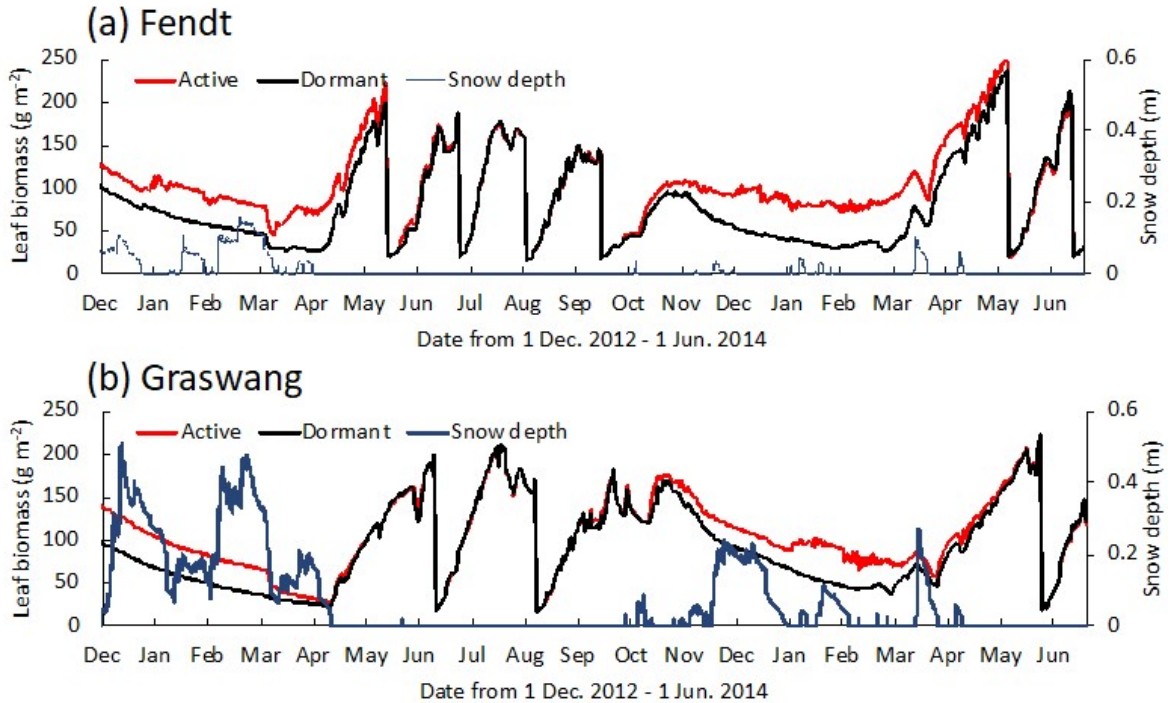

**Figure 5.** Time series for calculated lea f biomass and snow depth (blue lines) at (a) Fendt and (b) Graswang from 1 December, 2012 until 1 June, 2014, in active (red lines) and dormant cases (black lines).

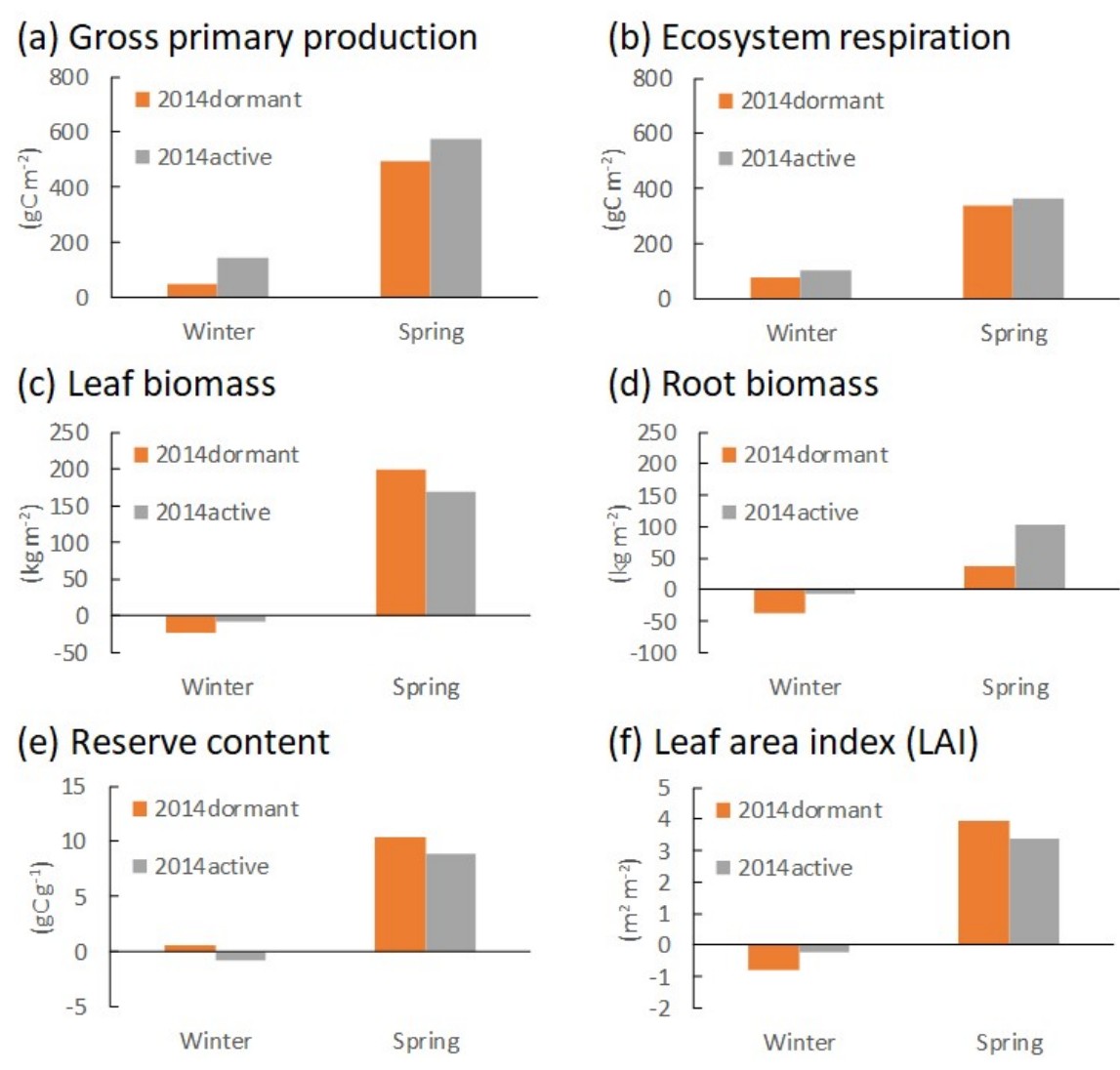

**Figure 6.** Changes in calculated (a) gross primary production (GPP), (b) ecosystem respiration, (c) live leaf and (d) root biomasses, (e) reserve content, and (f) leaf area index (LAI) at Fendt during the winter (from December to February) and spring (from March to May) in 2014 in active (red lines) and dormant cases (black lines).





**Table 1.** Characteristics of past or ongoing $CO_2$ flux observational sites over grassland ecosystems in European mountains. Snow-free $CO_2$ uptake (bold font) represents the situation of high negative values of $CO_2$ flux even during the wintertime (typically from December to February).

| Site name | Elevation (m) | MAT (°C) | MAP (mm) | Snow-free $CO_2$ uptake | Number of cuts per year | Source |
|---|---|---|---|---|---|---|
| Chamau | 393 | 9.8 | 1184 | **Yes** | 6-7 | Zeeman et al. (2010) |
| Oensingen | 452 | 9.5 | 1100 | **Yes** | 3 | Ammann et al. (2009) |
| Rotholz | 523 | 8.2 | 1151 | No | 3 + occasional grazing | Wohlfahrt et al. (2010) |
| Fendt | 600 | 8.0 | 1100 | **Yes** | 4-6 | Zeeman et al. (2017) |
| Rottenbuch | 760 | 8.0 | 1000 | **Yes** | 5 | Zeeman et al. (2017) |
| Graswang | 865 | 6.0 | 1000 | No | 2 | Zeeman et al. (2017) |
| Neustift | 970 | 6.3 | 852 | No | 3 | Wohlfahrt et al. (2008) |
| Frëbüel | 982 | 7.5 | 1708 | No | 4 | Zeeman et al. (2010) |
| Seebodenalp | 1025 | 7.3 | 1327 | No | 2 | Rogiers et al. (2005) |
| Dischma | 1250 | 2.8 | 1022 | No | 2 + occasional grazing | Merbold et al. (2013) |
| Monte Bondone | 1553 | 5.5 | 1189 | No | 1 | Marcolla et al. (2010) |
| Torgnon | 2160 | 3.1 | 880 | No | 0 | Galvagno et al. (2013) |



**Table 2.** Simulation settings for the modified SOLVEG at Fendt and Graswang sites. Abbreviations: DM, dry matter; DW: dry weight.

| Items | Values | Key reference |
|---|---|---|
| Time step | 100 s | This study |
| Numbers of layers | 15, 8, and 7 for atmosphere, vegetation, and soil, respectively | This study |
| Soil layer boundaries | 0.02, 0.05, 0.1, 0.2, 0.5, 1.0, and 2.0 m depth | This study |
| Vegetation layer boundaries | 0.05-0.5 m height with an increment of 0.05 m | This study |
| Atmospheric layer boundaries | Vegetation layers and 0.6, 0.8, 1.2, 1.6, 2.0, and 4.0 m height | This study |
| Soil texture | Silt | This study |
| Porosity | $0.55 \text{ m}^3 \text{ m}^{-3}$ | This study |
| Initial and bottom soil temperature | 0 °C for all soil layers | This study |
| Snow layer thickness | 5 mm | This study |
| Empirical parameter, $C_k$ | 8 | Zhang et al. (2007) |
| Irreducible liquid water content in snow | $0.03 \text{ m}^3 \text{ } m^{-3}$ | Hirashima et al. (2010) |
| Other parameters for snow and soil frozen sub-model | | Same as Jordan (1991) |
| Maximum catalytic capacity of Rubisco at 25 °C | $45 \text{ } \mu \text{ mol m}^{-2} \text{ s}^{-1}$ | This study and within range of Wohlfahrt et al. (2001) |
| Dark respiration rate of leaves at 25 °C | $1.52 \text{ } \mu\text{mol m}^{-2} \text{ s}^{-1}$ | Wohlfahrt et al. (2001) |
| Activation energy for dark respiration | $48.9 \text{ kJ mol}^{-1}$ | Wohlfahrt et al. (2001) |
| Minimum stomatal conductance | $0.08 \text{ mol m}^{-2} \text{ s}^{-1}$ | Wohlfahrt et al. (2001) |
| Threshold air temperature when photosynthesis starts, $T_{ph}$ | 1 and 11 °C at Fendt and Graswang | This study |
| Other parameters for vegetation sub-model | | C3-grass (Nagai, 2004) |
| Initial leaf area index (LAI) | $1.5 \text{ m}^2 \text{ m}^{-2}$ | This study |
| Initial carbohydrate storage | $100 \text{ kgDM ha}^{-1}$ | This study |
| Initial root biomass | $7000 \text{ kgDM ha}^{-1}$ | This study |
| Initial total tiller density | $1000 \text{ number m}^{-2}$ | This study |
| Ratio of total generative tiller | 0.1 | Höglind et al. (2016) |
| Ratio of fast generative tiller | 1.0 | Höglind et al. (2016) |
| Initial total tiller density | $1000 \text{ number m}^{-2}$ | This study |
| Initial stem biomass | $0 \text{ kgDM ha}^{-1}$ | This study |
| Initial stubble biomass | $0 \text{ kgDM ha}^{-1}$ | This study |
| Initial specific leaf area (SLA) | $0.002 \text{ m}_2 \text{ kgDW}^{-1}$ | This study |
| Maximum SLA | $0.003 \text{ m}^2 \text{ kgDW}^{-1}$ | Zeeman et al. (2017) |
| LAI after the grass cut | $0.5 \text{ m}^2 \text{ m}^{-2}$ | This study |
| Root life span (residence time) | $0.001 \text{ d}^{-1}$ (2.74 yr) | Höglind et al. (2016) |
| Other parameters related to BASGRA module | | Same as Höglind et al. (2016) |
| Parameters for soil microbiological processes | | Same as Ota et al. (2013) |





**Table 3.** Comparisons of daily values between SOLVEG simulations and observations at Fendt and Graswang sites from 1 December, 2011, to 1 November, 2014. $R_{net}$, net radiation; $H$, sensible heat flux; $\lambda E$, latent heat fluxes; $F_{CO2}$, CO$_2$ flux. ME, average error = simulations - observations; RMSE, root-mean-square error.

| Variables | Fendt (595 m asl.) | | | | | Graswang (864 m asl.) | | | | |
|---|---|---|---|---|---|---|---|---|---|---|
| | ME | RMSE | Intercept | Slope | R | ME | RMSE | Intercept | Slope | R |
| $R_{net}$ (W m$^{-2}$) | 0.130 | 10.139 | 3.161 | 0.946 | 0.987 | -3.114 | 11.486 | 5.979 | 0.940 | 0.984 |
| $H$ (W m$^{-2}$) | -4.268 | 18.354 | 5.093 | 0.920 | 0.781 | -6.105 | 24.783 | 7.114 | 0.889 | 0.747 |
| $\lambda E$ (W m$^{-2}$) | -4.112 | 31.471 | 17.875 | 0.819 | 0.910 | -3.145 | 32.770 | 17.801 | 0.816 | 0.905 |
| $F_{CO2}$ ($\mu$ mol m$^{-2}$ s$^{-1}$) | 0.579 | 3.372 | -1.353 | 0.788 | 0.785 | -0.133 | 3.372 | -0.712 | 0.838 | 0.837 |