# Peer review of "Wintertime grassland dynamics may influence below-ground biomass under climate change: a model analysis"

_Biogeosciences, 2019_

## Referee Comment (RC1) · Anonymous Referee #1 · 9 Oct 2019

General Comments: Overall, the paper does an excellent job using a process-based model to look at a critical period associated with montane grassland plants, the winter. Grasslands store lots of carbon belowground as roots and create a rich OM soil layer. The authors build a good argument for why they want to look at carbon fluxes and allocation during the winter months. The presentation of model results is good, and their conclusions adhere to what was found in their results. I do not have any major concerns with this paper as it stands.

Specific Comments: The authors discuss low temperature photosynthesis in both the introduction and conclusion but do not go further in depth about temperature thresh-

olds other than 5C when rubisco is very limited by temperature. I would suggest that the authors give a little more in the introduction about cold stress dynamics in relation to rubisco. The second aspect of the carbon dynamics that should be addressed is how water movement is impacting photosynthesis and carbon allocation within the grassland at these low temperatures. The dynamics associated between photosynthesis and water need to be stated, especially when discussing freezing conditions that occur during winter. The last specific comment I have is that the authors talk about the grasslands as fodder for livestock and its importance in the introduction, but the authors do not revisit this broader impact in the discussion.

Technical Corrections: Pg 2 lines 25-27 – This is an awkward sentence. Pg 2 line 33 – I do not understand what is meant by "... the above change in snow cover conditions...", please state what changes in snow cover conditions, make the readers job easy to remember conditions or treatments. Pg 8 lines 220-225 – Please look at Sage and Kubien 2007 Plant, Cell and Environment. This article discusses how temperature influences Rubisco, maybe a useful article for refence to help. Figure 1 – This is an extremely complicated figure and hard to understand. This figure might be better suited as a supplementary figure. To help improve clarity of the figure I would suggest decomposing the figure into easier to understand panels. For instance, maybe have one panel that focuses on atmospheric parameters, another on plant processes, and another on soil processes. I do understand that many of the processes are inter-connected. Figure 3 – The choice of having red and green on same figure is not color blind friendly. If one of the colors could be changed to a color-blind friendly palette that would enhance the clarity of the figure for all readers. Figure 6 – When printed in black and white the two colors orange and grey are too similar, please darken the grey to create a greater contrast between the two for improved interpretability when printed.

---

## Referee Comment (RC2) · Anonymous Referee #2 · 11 Oct 2019

General Comments: In general, the idea to look at wintertime carbon dynamics in grasslands and to include (biological) processes considering also cold stress in process-based models is very important. However, in my opinion this study doesn't make full use of this potential and is not very innovative. The 'new' model is based on integrating an already existing grass growth model into the SOLVEG model. However, whether this leads to an improvement or not is not analyzed (comparison of SOLVEG with and without the coupling of BASGRA) although it seems like relevant observational data exist to evaluate model performance. The model is run with pre-determined parameters and running the model under two scenarios, where one is determined to more or less shut down grass physiology and photosynthesis during low temperatures,

leads in this case – unsurprisingly – to less carbon uptake /productivity during wintertime. While the outcome for the other processes might be more interesting, they are simply the result of how the model is set-up in the first place... At least, this is discussed to some extent in the last paragraph.

Evaluating that the underlying processes of the model actually represent the 'true' ecosystem processes and calibrating the model parameters using an optimization process and the observational data, the model could be used to investigate differences between the study sites or to analyze different (climate) scenarios.

Specific Comments:

Line 65: 'Observational data are used...' – what data exactly? How is it obtained/measured?

Line 121 ff, 2.2 Empirical parameterizations for cold acclimation: why not use an optimization algorithm to determine the parameter(s) for the two sites and compare if there are significant differences?

Line 154/155: 'Since the lack of the data, most of the micrometeorological and hydrological parameters for SOLVEG runs are from previous studies conducted at the study sites' Lack of what kind of data? Studies from previous years? Are those comparable to the actual years?

Line 158/159: 'The unknown parameter, the threshold temperature for cold stresses [Tph in Eq. (2)], is manually determined' – based on which values? Are different values somehow compared and evaluated?

Line 187/188: it seems like that at Graswang there was more snow than in Fendt even during the warm winter. In general, are the climatic conditions comparable at the two sites?

Line 200: what about the results from Graswang?

Line 202: 100 gC m-2 – cumulative? per year? Per season?

Line 202: changes in leaf biomass in spring in the active scenario were lower because the starting leaf biomass was higher

Line 210/211: 'In particular, the model reproduced the low or near-zero $CO_2$ uptake during the normal winter period at the Graswang, regulated by the lowering of soil temperature due to snowfall'. In line 99 you write: 'When snow covers grasses, no photosynthesis is assumed to occur...' so this might actually (also) be a reason for the observed model output!!

Line 218: 'high $CO_2$ uptake rate at low altitude during winter was likely explained by high levels of physiological activity of grasslands' – what else could be the reason? Isn't the interesting question what influences physiological activity of grasslands during winter?

Line 222-226 this whole paragraph is not written clearly. Line 223: 'The study site' – which study site, Fendt or Graswang?

Line 230: 'annual mean air temperature (MAT) of less than 8 °C' should be 'more than 8 °C'

Line 247: 'increased annual GPP to 100 gC m$-2$': from Fig. 6 it seems higher, the bar just for spring shows already about 600 gC m-2 (which seems quite much for a grassland)

Line 247f: 'increased annual GPP to 100 gC m$-2$ due to cold acclimation in the active scenario': before you write that acclimation leads to a decline in photosynthetic capacity (see line 224-225).

Line 248f: 'but the above-ground biomass of the first cutting simulated in this scenario was less than that in the dormant scenario' – actually, in Fig. 5 you show that leaf biomass in the active scenario at the first cutting is higher or as high as in the dormant scenario... The change from March to May is smaller, because the value in March is

already much higher for the active scenario... See also comment to Line 202.

Fig. 4: So the results for Fendt are always with Tph = 1 °C and for Graswang Tph = 11 °C? How do the results look for the respective other simulation? Table 3 is more or less a repetition of Fig. 4; ME and RMSE could also be included in Fig. 4.

Supplement: 'Modeling grassland vegetation growth and development' – the whole paragraph contains a lot of repetition of what is already written in the main article's Material and Methods section.

Technical corrections:

Line 26: 'to a low' delete 'a'

Line 27: delete 'being dormant'

Line 33: 'it is necessary to understand the response of grassland productivity to changes in snow cover conditions in order to...'

Line 37: delete 'of winter stress'

Line 75: 'are' instead of 'were'

Line 83: delete 'either'

Line 91: 'consists of both'

Line 130: 'leaves' instead of 'leaf'

Line 131: 'layers'

Line 140: 'of the sites'

Line 141: delete ', which is an area'

Line 147: $H_2O$ subscript

Line 158: 'The threshold temperature for cold stresses...'

Throughout the manuscript: 'study period' instead of 'simulation/calculation period'

Fig. 2: check panel labels

Fig. 3: check panel labels in the caption.

Fig. 4: Adjust axis min/max to the min/max of the data. Check panel labels (2 times panel d) )

Fig. 5: leaf not lea f

Fig. 6: red lines? Black lines? Should be grey bars / orange bars

---

## Author Comment (AC1) · 9 Nov 2019

Please see the attached zip file.

Please also note the supplement to this comment:
https://www.biogeosciences-discuss.net/bg-2019-305/bg-2019-305-AC1-supplement.zip

---

## Author Comment (AC2) · 9 Nov 2019

Please see the attached zip file.

Please also note the supplement to this comment:
https://www.biogeosciences-discuss.net/bg-2019-305/bg-2019-305-AC2-supplement.zip

―――――――――――――――――

---

## Author Response (AR1)

**Title: Wintertime carbon uptake of managed temperate grassland ecosystems may influence grassland dynamics**
**Authors: G. Katata et al.**

**Author response to reviewer comments**

**Response to Anonymous Referee #1**

◯General Comments: Overall, the paper does an excellent job using a process-based model to look at a critical period associated with montane grassland plants, the winter. Grasslands store lots of carbon belowground as roots and create a rich OM soil layer. The authors build a good argument for why they want to look at carbon fluxes and allocation during the winter months. The presentation of model results is good, and their conclusions adhere to what was found in their results. I do not have any major concerns with this paper as it stands.

*Response:* We appreciate your giving positive comments on our manuscript. We revised the manuscript based on your specific comments as follows. We hope that the manuscript is now ready to be published.

Specific Comments:
◯The authors discuss low temperature photosynthesis in both the introduction and conclusion but do not go further in depth about temperature thresholds other than 5C when rubisco is very limited by temperature. I would suggest that the authors give a little more in the introduction about cold stress dynamics in relation to rubisco.

*Response:* We admit the description of physiological processes and acclimation dynamics to cold stress was insufficient. To emphasize the physiological importance of our work, we added the sentences with relevant publications into Introduction and Discussion as: "As reviewed in Sage and Kubien (2007), most C3 plants show an increase in photosynthetic rate below the thermal optimum (cooler temperature) due to cold acclimation, associated with enhancements of starch and sucrose synthesis, electron transport capacity, and Rubisco content." (l.38-40) and "In our simulations, we treated these acclimation responses as a parameter change, although in future developments they might be described mechanistically in dependence on temperature development (Kumarathunge et al. 2019; Mediavilla et al. 2016). Other mechanisms are however, already implicitly considered in the photosynthesis model. For example, the limitation of photosynthesis and thus the optimum temperature shifts under low air temperature from electron-transport limited to Rubisco-limited (Sage and Kubien, 2007)." (l.242-246).

Kumarathunge, D. P., Medlyn, B. E., Drake, J. E., Tjoelker, M. G., Aspinwall, M. J., Battaglia, M., et al. (2019). Acclimation and adaptation components of the temperature dependence of plant photosynthesis at the global scale. *New Phytol.* 222, 768-784. doi: 10.1111/nph.15668
Mediavilla, S., González-Zurdo, P., Babiano, J. and Escudero, A. (2016). Responses of photosynthetic parameters

to differences in winter temperatures throughout a temperature gradient in two evergreen tree species. *Eur. J. Forest Res.* 135, 871-883. doi: 10.1007/s10342-016-0980-9

Sage, R. F. and Kubien, D. S. (2007). The temperature response of C3 and C4 photosynthesis. *Plant Cell Environ.* 30, 1086-1106. doi: 10.1111/j.1365-3040.2007.01682.x

○The second aspect of the carbon dynamics that should be addressed is how water movement is impacting photosynthesis and carbon allocation within the grassland at these low temperatures. The dynamics associated between photosynthesis and water need to be stated, especially when discussing freezing conditions that occur during winter.

*Response:* It is correct that drought as well as freezing could influence photosynthesis and growth behavior of plants. However, drought does not play a major role at our sites which has been explicitly addressed as "No drought stress to grasslands was apparent in the simulations at both sites during the study period (not shown in the figure)." (l.195-196).

○The last specific comment I have is that the authors talk about the grasslands as fodder for livestock and its importance in the introduction, but the authors do not revisit this broader impact in the discussion.

*Response:* As you suggested, we added a paragraph to the discussion that points to the importance of wintertime carbon uptake for livestock: "Therefore, the increased photosynthesis in the warmer winter does not necessarily increase grass yields, and thus fodder in mountainous regions. In order to quantify the impact on livestock supply, further research needs to investigate to which degree additional biomass is directed into above- and below-ground storages." (l.277-279).

Technical Corrections:

○Pg 2 lines 25-27 – This is an awkward sentence.

*Response:* We corrected the sentence as "For example, winter conditions that are characterized by low temperature limits the productivity of grassland vegetation either directly due to its effects on photosynthesis or indirectly by inducing senescence and dormancy, particularly at high elevation areas." (l.26-28).

○Pg 2 line 33 – I do not understand what is meant by ": : : the above change in snow cover conditions: : :", please state what changes in snow cover conditions, make the readers job easy to remember conditions or treatments.

*Response:* We acknowledge the complaint and revised the sentence as "These differences between grassland sites at different altitudes clearly indicate the importance of considering the responses to environmental changes that are expected under climate change. This particularly refers to the snow-free winter periods that affect air and soil temperatures and thus the whole carbon cycle in mountain grassland ecosystems." (l.34-36).

○Pg 8 lines 220-225 – Please look at Sage and Kubien 2007 Plant, Cell and Environment. This article discusses how temperature influences Rubisco, maybe a useful article for refence to help.

*Response:* Thank you so much for useful information. We considered this information in the introduction and discussion sections as mentioned above (l.242-246).

◯Figure 1 – This is an extremely complicated figure and hard to understand. This figure might be better suited as a supplementary figure. To help improve clarity of the figure I would suggest decomposing the figure into easier to understand panels. For instance, maybe have one panel that focuses on atmospheric parameters, another on plant processes, and another on soil processes. I do understand that many of the processes are inter-connected.
*Response:* As you suggested, Fig. 1 was moved to the supplement (Fig. S1). Furthermore, the figure was decomposed to four panels for each submodel(s) as reader-friendly.

◯Figure 3 – The choice of having red and green on same figure is not color blind friendly. If one of the colors could be changed to a color-blind friendly palette that would enhance the clarity of the figure for all readers.
*Response:* The color of triangles in old Fig. 3 (now Fig. 2) was changed from green to orange as you suggested.

◯Figure 6 – When printed in black and white the two colors orange and grey are too similar, please darken the grey to create a greater contrast between the two for improved interpretability when printed.
*Response:* The colors in old Fig. 6 (now Fig. 5) were revised for the black and white colored style.

**Title: Wintertime carbon uptake of managed temperate grassland ecosystems may influence grassland dynamics**
**Authors: G. Katata et al.**

**Author response to reviewer comments**

**Response to Anonymous Referee #2**

〇General Comments: In general, the idea to look at wintertime carbon dynamics in grasslands and to include (biological) processes considering also cold stress in process-based models is very important.

*Response:* We appreciate your comment on the importance of our motivation. All of your suggestions were very crucial for improving our manuscript. After all revisions were made in the manuscript as follows, we hope that it is substantially improved and is now ready for publication.

〇However, in my opinion this study doesn't make full use of this potential and is not very innovative. The 'new' model is based on integrating an already existing grass growth model into the SOLVEG model. However, whether this leads to an improvement or not is not analyzed (comparison of SOLVEG with and without the coupling of BASGRA) although it seems like relevant observational data exist to evaluate model performance.

*Response:* We are presenting an integrated analysis that includes an enhancement of modelling abilities because the original model SOLVEG treated vegetation properties as prescribed, while the combination with BASGRA enables a dynamic, process-based representation of vegetation dynamics. If we would drive SOLVEG with measured LAI and biomass, it would necessarily perform better regarding gas exchange, but long-term effects due to seasonal variations in carbon allocation could not be detected. Since uncovering this effect is our main intention in this manuscript, a comparison to the uncoupled version that uses the evaluation data as input doesn't seem appropriate. In order to clarify this point and minimize misunderstandings, we modified several sentences that imply a "model development" instead of a model coupling in Abstract and Introduction. Also, the following sentences were added to the "Calibration and validation procedure" subsection: "Direct comparisons between the results using the original (SOLVEG only) and integrated models (SOLVEG coupled with BASGLA) are difficult because the vegetation dynamics had been prescribed in the original model, requiring time series of total LAI or leaf biomass data, which is used for evaluation in this study. Thus, we simply focus on the calibration of the integrated model only to investigate the impact of wintertime carbon uptake on grassland dynamics." (l.157-160).

〇The model is run with pre-determined parameters and running the model under two scenarios, where one is determined to more or less shut down grass physiology and photosynthesis during low temperatures, leads in this case – unsurprisingly – to less carbon uptake /productivity during wintertime. While the outcome for the other processes might be more interesting, they are simply the result of how the model is set-up in the first

place … At least, this is discussed to some extent in the last paragraph.

*Response:* Thank you for your suggestion to improve the discussion. We now try to link the different model settings better to actual physiological processes. Therefore, we added the following sentences and refer to additional relevant publications in Introduction and Discussion as: "As reviewed in Sage and Kubien (2007), most C3 plants show an increase in photosynthetic rate below the thermal optimum (cooler temperature) due to cold acclimation, associated with enhancements of starch and sucrose synthesis, electron transport capacity, and Rubisco content." (l.38-40) and "In our simulations, we treated these acclimation responses as a parameter change, although in future developments they might be described mechanistically in dependence on temperature development (Kumarathunge et al. 2019; Mediavilla et al. 2016). Other mechanisms are however, already implicitly considered in the photosynthesis model. For example, the limitation of photosynthesis and thus the optimum temperature shifts under low air temperature from electron-transport limited to Rubisco-limited (Sage and Kubien, 2007)." (l.242-246).

Kumarathunge, D. P., Medlyn, B. E., Drake, J. E., Tjoelker, M. G., Aspinwall, M. J., Battaglia, M., et al. (2019). Acclimation and adaptation components of the temperature dependence of plant photosynthesis at the global scale. *New Phytol.* 222, 768-784. doi: 10.1111/nph.15668

Mediavilla, S., González-Zurdo, P., Babiano, J. and Escudero, A. (2016). Responses of photosynthetic parameters to differences in winter temperatures throughout a temperature gradient in two evergreen tree species. *Eur. J. Forest Res.* 135, 871-883. doi: 10.1007/s10342-016-0980-9

Sage, R. F. and Kubien, D. S. (2007). The temperature response of C3 and C4 photosynthesis. *Plant Cell Environ.* 30, 1086-1106. doi: 10.1111/j.1365-3040.2007.01682.x

○Evaluating that the underlying processes of the model actually represent the 'true' ecosystem processes and calibrating the model parameters using an optimization process and the observational data, the model could be used to investigate differences between the study sites or to analyze different (climate) scenarios.

*Response:* Thank you for this comment. We fully that these issues are important and could be addressed with the presented methodology in the future. To highlight this opportunity, the following sentences were added to the revised manuscript: "The presented approach and model combination could be used in the future for analyzing climate change scenarios and the site dependency of responses. This will require more comprehensive datasets for evaluation, with which the importance of underlying processes can be revealed and model calibration can be carried out, possibly using an optimization procedure such as Monte Carlo simulation (e.g., Van Oijen et al., 2005)." (l.229-232).

Specific Comments:
○Line 65: 'Observational data are used …' – what data exactly? How is it obtained/measured?
*Response:* We apologize for the uncomprehensive sentence; we revised as follows: "At the upper boundary conditions, the variables of horizontal wind speeds, potential temperature, specific humidity (and liquid water

content of the fog, gas and aerosol concentrations, if available) are typically obtained from hourly or half-hourly observational data. For further explanations see section 2.3." (l.68-71).

○Line 121 ff, 2.2 Empirical parameterizations for cold acclimation: why not use an optimization algorithm to determine the parameter(s) for the two sites and compare if there are significant differences?

*Response:* It is true that using optimization procedures for model parametrization enables a more objective picture that could be used for site comparison and identification of process deficiencies. However, we feel that the current data set is too limited for such an exercise and holds some pitfalls in case of this coupled model exercise. Nevertheless, the possibility of using this methodology has now been addressed as already mentioned in the answers to general comments (l. 229-232).

○Line 154/155: 'Since the lack of the data, most of the micrometeorological and hydrological parameters for SOLVEG runs are from previous studies conducted at the study sites' Lack of what kind of data? Studies from previous years? Are those comparable to the actual years?

*Response:* The sentence was inappropriate. We revised it as "Typical values of soil hydrological parameters (e.g., saturated hydraulic conductivity) in the study area are given to SOLVEG runs from the past model study (Hingerl et al., 2016)." (l. 162-164).

○Line 158/159: 'The unknown parameter, the threshold temperature for cold stresses [Tph in Eq. (2)], is manually determined' – based on which values? Are different values somehow compared and evaluated?

*Response:* More explanation was required. We added the sentence as "By changing the $T_{ph}$ value from the range between 1 and 11 °C with an increment of 2 °C (not shown in the figure), we obtained the best results as $T_{ph} = 1$ °C and 11 °C for Graswang and Fendt, respectively." (l.168-170).

○Line 187/188: it seems like that at Graswang there was more snow than in Fendt even during the warm winter. In general, are the climatic conditions comparable at the two sites?

*Response:* As shown in Table 1, mean annual air temperature differed by 2 °C between the two sites due to the difference in altitude (Table 1, Mauder et al. 2005, Zeeman et al. 2017). Precipitation in general is relatively similar at both sites. Simulation results includes this difference as input of meteorological data, which is important for the differences in snow coverage.

○Line 200: what about the results from Graswang?

*Response:* We insert new Fig. 6 for Graswang and added the sentence of "We focus on the Fendt site for illustration of the effect (Fig. 5) because the differences between scenarios were small for all variables at the Grasswang site." (l.211-213).

○Line 202: 100 gC m-2 – cumulative? per year? Per season?

*Response:* We revised the sentence as "this was particularly apparent as cumulative GPP differed by a factor of

three or by approximately 100 gC m$^{-2}$ per year".

○Line 202: changes in leaf biomass in spring in the active scenario were lower because the starting leaf biomass was higher

*Response:* That is true. And what we wanted to state here is that we expected that the change would be similarly high because simulated carbon reserve contents (a potential of carbon allocation to the above-ground biomass) in both winter and spring were similar in the two simulation scenarios (Fig. 5e). Still there was the indicated difference. This is now better emphasized as follows: "Thus, we expected that the change in the above-ground biomass would be higher in the active scenario because simulated carbon reserve contents (a potential of carbon allocation to the above-ground biomass) in winter were similar in the two simulation scenarios (Fig. 5e). However, the above-ground biomass at the first cutting simulated in the active scenario was similar that in the dormant scenario (Fig. 5c)." (l.269-273).

○Line 210/211: 'In particular, the model reproduced the low or near-zero CO2 uptake during the normal winter period at the Graswang, regulated by the lowering of soil temperature due to snowfall'. In line 99 you write: 'When snow covers grasses, no photosynthesis is assumed to occur...' so this might actually (also) be a reason for the observed model output!!

*Response:* The sentence was inappropriate; it is now revised as "In particular, the model reproduced the low or near-zero CO2 uptake during the normal winter period at the Graswang as a response to low soil temperatures that limit photosynthesis even throughout the snow-free conditions (Fig. 2d)." (l.223-225).

Line 218: 'high CO2 uptake rate at low altitude during winter was likely explained by high levels of physiological activity of grasslands' – what else could be the reason? Isn't the interesting question what influences physiological activity of grasslands during winter?

*Response:* We agree that the indicated formulation is not very informative. This section has been changed as indicated in responses to the general comments where underlying physiological processes are now addressed (l.242-246).

○Line 222-226 this whole paragraph is not written clearly.

*Response:* We agree that the paragraph was vaguely formulated; it is now revised as "A possible explanation for the lesser photosynthesis is a rapid acclimation response of grasslands to decline in photosynthetic capacity after the exposure to freezing temperatures since (e.g., Huner et al., 1993; Kolari et al., 2007). In fact, the Graswang site was exposed to frost during the extremely warm winter in 2013-2014 (Zeeman et al., 2017), which may support the above explanation. In our simulations, we treated these acclimation responses as a parameter change, although in future developments they might be described mechanistically in dependence on temperature development (Kumarathunge et al. 2019; Mediavilla et al. 2016). Other mechanisms are however, already implicitly considered in the photosynthesis model. For example, the limitation of photosynthesis and thus the optimum temperature shifts under low air temperature from electron-transport limited to Rubisco-limited (Sage and Kubien, 2007). Further

observational work is required at various grassland ecosystems in order to evaluate this hypothesis." (l.239-247).

○Line 223: 'The study site' – which study site, Fendt or Graswang?

*Response:* We revised this as "The Graswang site" (l.241).

○Line 230: 'annual mean air temperature (MAT) of less than 8 _C' should be 'more than 8 _C'

*Response:* This was a mistake; we revised it as "more than 8 °C" (l.252).

○Line 247: 'increased annual GPP to 100 gC m-2': from Fig. 6 it seems higher, the bar just for spring shows already about 600 gC m-2 (which seems quite much for a grassland)

○Line 247f: 'increased annual GPP to 100 gC m-2 due to cold acclimation in the active scenario': before you write that acclimation leads to a decline in photosynthetic capacity (see line 224-225).

*Response:* The sentence was inappropriate; we revised it as "According to this study, CO2 uptake at the Fendt site, estimated as an annual GPP of 100 gC m-2 in 2013-2014 was mainly due to the higher wintertime photosynthetic rate in the active scenario." (l.268-269).

○Line 248f: 'but the above-ground biomass of the first cutting simulated in this scenario was less than that in the dormant scenario' – actually, in Fig. 5 you show that leaf biomass in the active scenario at the first cutting is higher or as high as in the dormant scenario … The change from March to May is smaller, because the value in March is already much higher for the active scenario… See also comment to Line 202.

*Response:* Your suggestion is correct. As already described in the responses above is that we wanted to state here was that we expected a similarly high change, because simulated carbon reserve contents (a potential of carbon allocation to the above-ground biomass) in both winter and spring were similar in the two simulation scenarios (Fig. 5e) but it was not. This is emphasized as "However, the above-ground biomass of the first cutting simulated in the two scenarios (Fig. 5c). Since this was not the case, we revised the statement accordingly (see answer to Line 202)

○Fig. 4: So the results for Fendt are always with Tph = 1 _C and for Graswang Tph =11 _C? How do the results look for the respective other simulation?

*Response:* Further explanation about manual calibration was required. As responded above, we added the sentence as "By changing the $T_{ph}$ value from the range between 1 and 11 °C with an increment of 2 C (not shown in the figure), we obtained the best results as $T_{ph}$ = 1 °C and 11 °C for Graswang and Fendt, respectively." (l.168-170).

○Table 3 is more or less a repetition of Fig. 4; ME and RMSE could also be included in Fig. 4.

*Response:* We delete the Table 3 and ME and RMSE were incorporated into Fig. 3 (old Fig. 4).

○Supplement: 'Modeling grassland vegetation growth and development' – the whole paragraph contains a lot of repetition of what is already written in the main article's Material and Methods section.

*Response:* We completely deleted the subsection of "Modeling grassland vegetation growth and development" from

the supplement to avoid its duplication.

Technical corrections:
○Line 26: 'to a low' delete 'a'

*Response:* We correct the sentence accordingly.

○Line 27: delete 'being dormant'

*Response:* The sentence has been removed after we responded to the other reviewer.

○Line 33: 'it is necessary to understand the response of grassland productivity to changes in snow cover conditions in order to…'

*Response:* The sentence was revised based on the suggestion from the other reviewer as follows: "Clearly, it is necessary to understand the response of grassland productivity to the snow-free winter and air and soil temperatures in order to…" (l.33-34)

○Line 37: delete 'of winter stress'

○Line 75: 'are' instead of 'were'

○Line 83: delete 'either'

○Line 91: 'consists of both'

○Line 130: 'leaves' instead of 'leaf'

○Line 131: 'layers'

○Line 140: 'of the sites'

○Line 141: delete ', which is an area'

○Line 147: $H_2O$ subscript

○Line 158: 'The threshold temperature for cold stresses…'

○Throughout the manuscript: 'study period' instead of 'simulation/calculation period'

*Response:* We correct the words and sentences according to your suggestions. We appreciate your careful checking.

○Fig. 2: check panel labels

○Fig. 3: check panel labels in the caption.

○Fig. 4: Adjust axis min/max to the min/max of the data. Check panel labels (2 times panel d) )

○Fig. 5: leaf not lea f

○Fig. 6: red lines? Black lines? Should be grey bars / orange bars

*Response:* All figures were revised as you suggested, while figure numbers were reduced because previous Fig. 1 was moved to Fig. S1. Thank you so much.

[revised manuscript text omitted]
$^{-1}$) | 0.579 | 3.372 | -1.353 | 0.788 | 0.785 | -0.133 | 3.372 | -0.712 | 0.838 | 0.837 |

---

## Author Response (AR2)

**Title: Wintertime grassland dynamics may influence below-ground biomass under climate change: a model analysis**
**Authors: G. Katata et al.**

**Author response to editor comment**

The Referee makes interesting points but I note that perceptions of scientific novelty are not a criterion for acceptance or rejection to Biogeosciences. Please consider the referee comments in a letter addressed to me and. Will adjudge if the revised manuscript is sufficient for publication in Biogeosciences.

*Response:* We appreciate your comment on the importance of our motivation. All of your suggestions were very crucial for improving our manuscript. After all revisions were made in the manuscript as follows, we hope that it is substantially improved and is now ready for publication.

**Title: Wintertime grassland dynamics may influence below-ground biomass under climate change: a model analysis**
**Authors: G. Katata et al.**

**Author response to reviewer comments**

**Response to Anonymous Referee #2**

○ 'Wintertime carbon uptake of managed temperate grassland ecosystems may influence grassland dynamics'– the influence of reduced snowcover and its effects on carbon uptake was already analyzed based on the same data that is presented here in Zeeman et al 2017. Although the current study focuses more on model development and simulations and the data is only used for model validation, I still think what you present is not substantially new or innovative. In response to my review comment you write that uncovering long-term effects due to seasonal variations in carbon allocation is your main intention. However, this is not clear from the title and there is no straightforward introduction.

*Response:* Indeed, we think that a certain novelty of this paper origins from the potential change of the belowground carbon pool in grasslands, particularly in cold environments that may be specifically affected by climate change. This does not only incorporate the long-term storage but also living belowground biomass, which is different from the previous work by Zeeman et al. (2017) and may serve to inspiring future model developments. However, the reviewer is correct in stating that the title and the introduction do not fully reflect our intention. Therefore, we modified the title into "Wintertime grassland dynamics may influence below-ground biomass under climate change: a model analysis" and the whole Introduction section. Now these better explain the importance of considering carbon dynamics during winter, and leads more straightforward to model requirements and set-up of the simulation study.

○Moreover, from the results you present you are drawing inconsistent and unsupported conclusions. For example, you write in L 212 'differences between scenarios were small for all variables at the Grasswang site'. The new Figure 6, however, shows that the relative increase of GPP for the active scenario compared to the dormant scenario is probably of the same order of magnitude at the Graswang site than at Fendt. That the absolute numbers are smaller can most likely be attributed to the fact, that Graswang is at an higher altitude and therefore has lower temperatures and more snowcover even during the warm winter.

*Response:* We admit that our wording in this section is inconsistent with the presented results because we failed to update the text according to the newest simulations. We revised the respective sentence to "In the following, we focus on Fendt for illustration of the scenario differences (Fig. 5) but would like to emphasize that the responses are similar at both sites. The differences in absolute values, especially a smaller LAI and less biomass are due to the generally cooler conditions at the Graswang site. " (L210-213, P.7).

○And on the other hand, if the differences between the different scenarios indeed are not significant, then your argument that calibrated values for Tph for each site are necessary is lapsed since the one that fits best for Fendt apparently fits for Graswang well enough.

*Response:* We admit that our wording in this section is to some degree inconsistent with the presented results as also commented in one of the previous responses. Nevertheless, tuning the Tph values has been still required for each site so that the model reproduced the measured $CO_2$ fluxes, indicating that this parameter is site-specific (as explained in subsection 2.2). This is emphasized at various places in the manuscript, e.g. by adding the sentences as "It should be noted that such difference in environmental conditions between both sites required the calibration of the value of Tph for each site (subsection 2.2)." (L213-214, P.7)

○Moreover, I do not agree with your conclusion, that 'a large fraction of carbon (CO2) gained by photosynthesis during winter was […] used to grow roots during the spring growth period' (L 263ff). Neither the dormant nor the active scenario in Fendt as well as in Graswang show an increase in the reserve content during winter (the active scenario even shows a reduction), so there is no carbon stored during winter that can be used for root growth?

*Response:* The sentence was inappropriate and caused the misunderstanding that the additional carbon gain during winter is stored and results in more growth during spring. However, we assume that the carbon is directly used for growth (mostly root growth and only to a very minor degree foliage growth) since soil temperatures were sufficiently high. We therefore revised the sentences to "The comparison between scenarios shows that root biomass clearly increased in the active compared to the dormant simulation during winter and spring (Fig. 5c and d) which can only be due to $CO_2$ gain by photosynthesis during this time (Fig. 5a). Note that this somewhat counter-intuitive results may be due to the inability of the model to grow specific storage organs that could later be emptied to growth other tissues. A differentiation, however, is not yet possible because respective observations are not available." (L263-267, P.9).

○Moreover, the increased root biomass in spring in the active scenario can also be observed in Graswang, although you argue previously that there is no relevant photosynthesis during winter.

*Response:* There is lack of detailed explanation in the last revision and caused your misunderstanding. Root biomass increased also at the Graswang site because CO2 uptake occurred in active scenarios as shown in Fig. 6a. Thus, we deleted the word of "or near zero" from and inserted "with calibrated Tph value" into L225-226 in P.8, and also added the following sentences "It is obvious that leaf biomass during winter is higher in the active scenarios, mostly because of a higher leaf growth at the end of the vegetation period. In addition, some minor leaf growth also occurs in the snow free winter periods. Nevertheless, leaf biomasses converge during spring and are similar again at the first cutting event in May/June (Fig. 4b)." (L204-207, P.7).

○Indeed it is interesting to see, that the above ground biomass is almost the same at the first cutting for both scenarios. However, it seems like the rate of biomass gain at the beginning of spring is very high and about the same for both scenarios but levels off later indicating that a possible maximum is reached in both cases (e.g. no further increase due to light limitation when the canopy gets too dense). This could possibly be analyzed in more detail with different model scenarios…

*Response:* We appreciate your interest on our result and your suggestion. However, we feel that further scenario runs are not needed to discuss the phenomenon in more detail. To address this point, we added a paragraph that runs as follows: "Thus, it could be assumed that the increase in the above-ground biomass in spring would be higher in the active scenario. However, the above-ground biomass at the first cutting simulated in the active scenario was similar that in the dormant scenario (Fig. 5c). Still, the behavior is consistent with the simulated carbon reserve contents (a potential of carbon allocation to the above-ground biomass) in winter, which were similar in the two simulation scenarios (Fig. 5e). The actual limitation might have internal (e.g. determined growth) or external causes. For example, self-shading could result in decreasing carbon gain efficiency of new leaves which might induce a growth stop or an increase in senescence when the canopy gets denser. Indeed, calculated LAI values were similar to critical ones for self-shading shortly before the first cutting event in 2014 (Fig. 4), which is however not a process considered in the model and is thus the result of a reasonably parametrized determined growth. This is corroborated by a similar degree of leaf senescence in both scenarios (6.9 and -0.7 % at Fendt and Graswang, respectively)." (L271-279, P.9).

○If you want to improve and re-submit the manuscript I recommend as previously mentioned to carefully evaluate the underlying processes of the model and validate they actually present the 'true' ecosystem processes. Analyze the model output carefully and make sure your conclusions are consistent and that there is a clear central theme from the title, introduction, results to the discussion and conclusions.

*Response:* We acknowledge the recommendation and recognized that the manuscript indeed still lacked some consistency in several parts. We thus followed the suggestion to elaborate on the making title, introduction and result/discussion sections more consistent and straightforward.

○Make sure to cite the original papers. For example, you write: "but growth stops if soil temperatures are lower than 5 ∘C (Körner, 2008). […] In this situation, organic matter (organic carbon) produced by photosynthesis is not used for grass growth but accumulates in the plant as reserves during winter (e.g., Körner, 2008)." But Körner 2008 mentions the 5 °C limit and that organic matter accumulates in reserves, but this is not the original research and he refers to other papers.

*Response:* As suggested by the reviewer, the original paper of Rabenhorst (2005) and Hoch and Körner (2003) were

referred for growth stops and organic matter accumulates, respectively (L.41 and 43, P.2). We also checked all references in the manuscript to appropriately support our sentences.

0394Δ 8710Δ 0331 014B 2032'

[revised manuscript text omitted]